# Development of a questionnaire to evaluate patients' awareness of cardiovascular disease risk in England's National Health Service Health Check preventive cardiovascular programme

Maria Woringer,[1] Jessica Jones Nielsen,[2] Lara Zibarras,[2] Julie Evason,[3] Angelos P Kassianos,[4] Matthew Harris,[5] Azeem Majeed,[1] Michael Soljak[1]

Presented at the NHS Health Check 2015 – Improvement through Collaboration conference and the First International Conference of Public Health, Primary Care and Congress of Person Centred Medicine.

For numbered affiliations see end of article.

**Correspondence to**
Dr Maria Woringer;
m.woringer@imperial.ac.uk

## ABSTRACT

**Background** The National Health Service (NHS) Health Check is a cardiovascular disease (CVD) risk assessment and management programme in England aiming to increase CVD risk awareness among people at increased risk of CVD. There is no tool to assess the effectiveness of the programme in communicating CVD risk to patients.

**Aims** The aim of this paper was to develop a questionnaire examining patients' CVD risk awareness for use in health service research evaluations of the NHS Health Check programme.

**Methods** We developed an 85-item questionnaire to determine patients' views of their risk of CVD. The questionnaire was based on a review of the relevant literature. After review by an expert panel and focus group discussion, 22 items were dropped and 2 new items were added. The resulting 65-item questionnaire with satisfactory content validity (content validity indices≥0.80) and face validity was tested on 110 NHS Health Check attendees in primary care in a cross-sectional study between 21 May 2014 and 28 July 2014.

**Results** Following analyses of data, we reduced the questionnaire from 65 to 26 items. The 26-item questionnaire constitutes four scales: Knowledge of CVD Risk and Prevention, Perceived Risk of Heart Attack/Stroke, Perceived Benefits and Intention to Change Behaviour and Healthy Eating Intentions. Perceived Risk (Cronbach's $\alpha$=0.85) and Perceived Benefits and Intention to Change Behaviour (Cronbach's $\alpha$=0.82) have satisfactory reliability (Cronbach's $\alpha$≥0.70). Healthy Eating Intentions (Cronbach's $\alpha$=0.56) is below minimum threshold for reliability but acceptable for a three-item scale.

**Conclusions** The resulting questionnaire, with satisfactory reliability and validity, may be used in assessing patients' awareness of CVD risk among NHS Health Check attendees.

## INTRODUCTION

Cardiovascular disease (CVD) is a major cause of disability and premature mortality worldwide. In England, it accounts for a third of deaths and costs the National Health Service

### Strengths and limitations of this study

► Questionnaire guided by literature review, expert panel, patient focus group and data analysis.
► Largely developed among 110 individuals representative of the target population.
► Face validity assessed via a patient focus group not representative of the target population.

(NHS) and the UK economy £30 billion annually.[1 2] Modifiable lifestyle risk factors, associated with 90% of CVD,[3 4] contributed to only 34% of the overall decline in CVD mortality in England between 2000 and 2007.[5] In 2010/2011, there were 1.4 million CVD-related hospital admissions, of which 60% were for people younger than 75 and more than half as an emergency. Further gains could be made in preventing long-term illness and disability associated with CVD while reducing healthcare costs by promoting healthier lifestyle changes.[6]

The NHS Health Check programme may be important for preventing premature CVD while reducing healthcare costs therein by identifying individuals at increased risk of CVD, raising their awareness of CVD risk and helping them manage their risk.[7–10] This CVD risk assessment and management programme was launched by the Department of Health in April 2009 in England among those aged 40–74 years free of vascular disease diagnosis.[7] It aims to prevent heart disease, stroke, diabetes and kidney disease while reducing health inequalities. Individuals' sociodemographics, cholesterol, blood pressure, smoking and family history of CVD are used to predict CVD risk.[11] In addition to lifestyle advice given to all participants, people at high risk of CVD are invited for

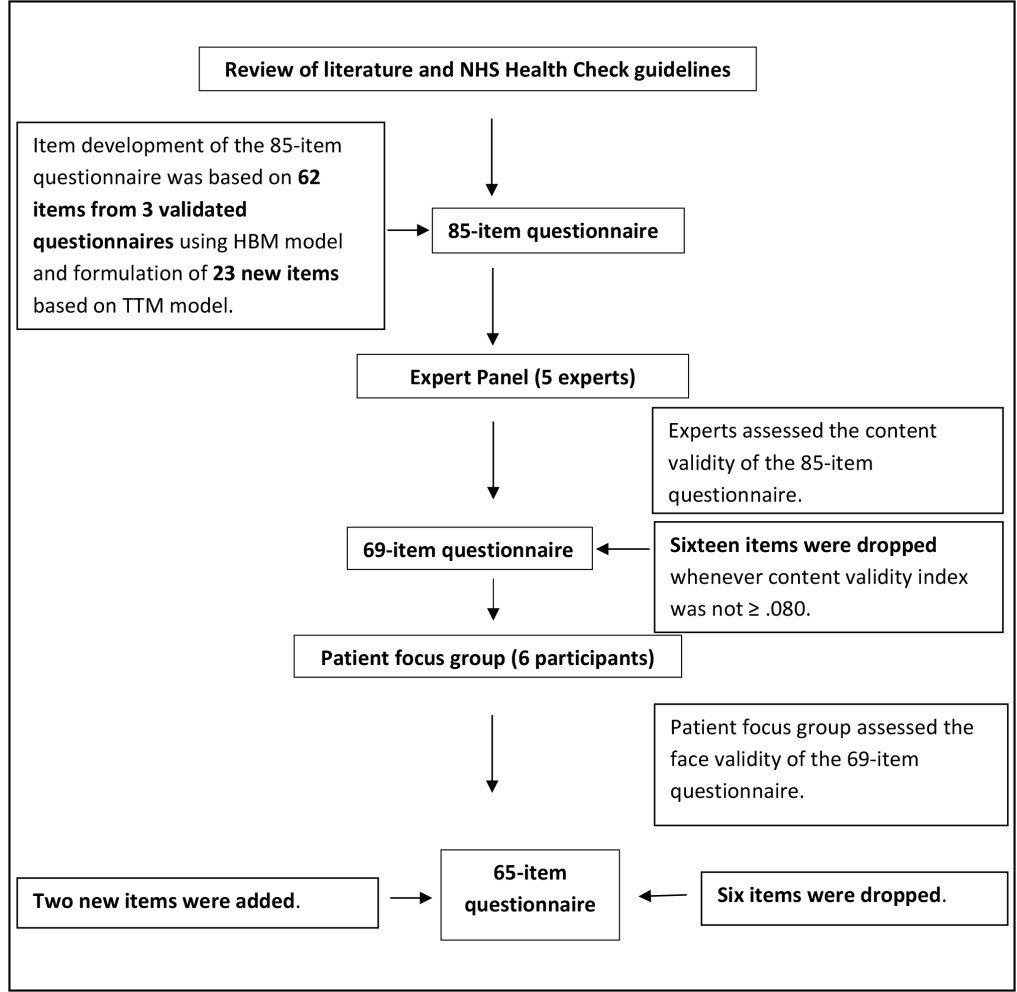

**Figure 1**  Flow chart of phase I of questionnaire development. HBM, Health Belief Model; NHS, National Health Service; TTM, Transtheoretical Model.

further consultations and offered statins and behaviour change support in relation to physical activity, smoking cessation, safe alcohol consumption and healthy diet. Projected programme cost is £180–£243 million/year with estimated cost per quality adjusted life-year (QALY) at £3000.1[10]

To adopt healthy lifestyle behaviours related to diet, exercise, smoking and alcohol consumption, the general population must be aware of CVD risk.[12] In the context of the NHS Health Check Programme, CVD risk awareness refers to the accuracy of perceived risk of CVD against predicted CVD risk, general knowledge of CVD and what one can do to lower predicted CVD risk. Whereas predicted CVD risk refers to one's chance of experiencing a heart attack or stroke,[11] perceived risk of CVD refers to a person's perception of their CVD risk. While as many as 40% of the general population underestimate their CVD risk, 20% overestimate their risk.[13] False reassurance may lead to adoption and/or maintenance of unhealthy behaviours contributing to the premature onset of CVD. Low CVD risk awareness is reported among men, inner city residents and people of lower socioeconomic

status.[12 14 15] It is not known if the Health Check results in improved CVD risk awareness.

Although several validated questionnaires measure knowledge, perceptions of CVD or intention to change behaviour,[15–17] no short, validated questionnaire assesses CVD risk awareness using all of these scales. Until now studies examining the accuracy of perceived risk and knowledge of CVD relied on non-validated tools.[16] The problem with using non-validated tools is that the questions may not accurately and reliably capture individuals' views or measure what they intend to measure. The aim of this work was to develop a questionnaire with satisfactory face, content validity and reliability to assess patients' awareness of CVD risk among NHS Health Check attendees.

## METHODS

The first phase of development of the questionnaire was guided by a literature review, an expert panel and a patient focus group. At each stage of questionnaire development, the number of items was reduced (see figure 1).

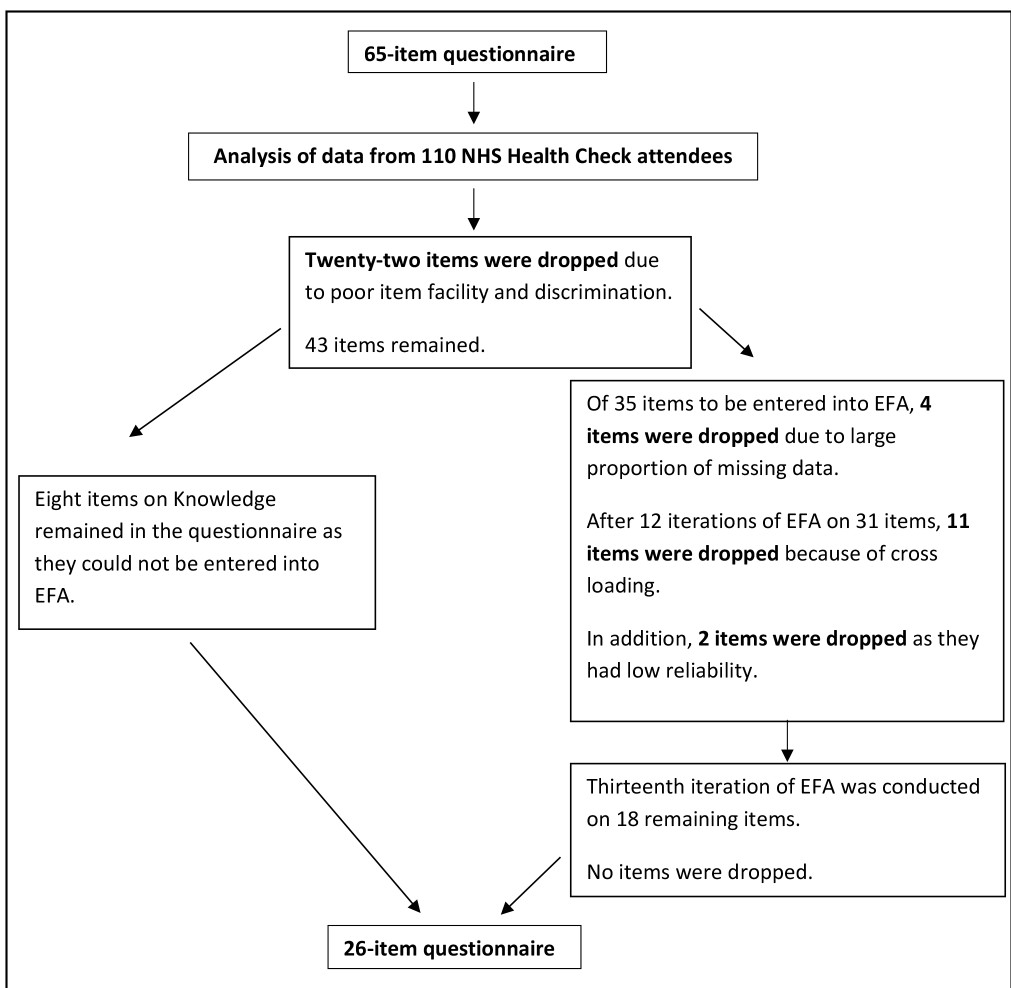

**Figure 2** Flow chart of phase II of questionnaire development. EFA, exploratory factor analysis; NHS, National Health Service.

The second phase of questionnaire development was guided by an analysis of data from 110 NHS Health Check attendees who completed the 65-item questionnaire. The number of questionnaire items was further reduced (see figure 2).

### Phase I of questionnaire development
#### Construction of draft questionnaire by review of relevant literature
We performed an extensive literature review pertaining to CVD risk awareness between December 2013 and January 2014 in the areas of disease knowledge, risk perception, intention to change and self-efficacy related to CVD and the Health Belief Model (HBM) to guide initial item development. PubMed and PsycINFO databases and Google Scholar Articles were used to search for existing instruments that measure perception of CVD risk, CVD knowledge and self-efficacy with no limits on the year of publication. The following keywords were used to identify the relevant literature: 'cardiovascular disease', 'heart disease', 'knowledge', 'risk', 'test', 'questionnaire', 'scale', 'assessment', 'self-efficacy', 'perception', 'health belief model'. Questionnaires were considered if they addressed CVD risk awareness, reported moderate-to-high scores of reliability and validity in population studies

and had suitable wording and level of understanding. Questionnaires were excluded if they pertained to individuals under the age of 15 as these people would not be eligible to receive an NHS Health Check, focused on risk unrelated to heart attack or stroke, and were not written in English.

Although a number of questionnaires were found measuring different aspects of CVD risk awareness such as heart disease knowledge, perception of CVD risk, perceived susceptibility and severity of CVD and benefits and barriers to adopting healthy behaviours,[17–19] no single questionnaire encompassed them all. Initial item development was guided by HBM[20] and the Transtheoretical Model (TTM).[21] According to HBM, individuals who have accurate knowledge of CVD and perceived susceptibility to and consequences of the disease, and are aware of the benefits of taking preventive measures are more likely to make important lifestyle choices to prevent the onset of disease.[22] The TTM describes behavioural change as a staged process over time including precontemplation, contemplation, preparation, action and maintenance.[21] Sixty-five items were selected using validated questionnaires addressing CVD knowledge,

and the main constructs of HBM such as perceived susceptibility, perceived severity, perceived benefits of changing behaviours and perceived barriers to making changes.[17–19] In addition, 23 new items were generated to identify perceived levels of readiness to engage in CVD risk reduction behaviours (using TTM) and self-efficacy (confidence in ability to change health behaviour) in relation to exercise, diet, smoking cessation and decreasing alcohol consumption.[23 24] These items were based on data collected during an NHS Health Check and behaviour-specific recommendations such as stopping smoking, consuming no more than 14 units of alcohol a week, eating at least five portions of fruit and vegetables a day and exercising at least 150 min per week.[25–28] The resulting 85-item questionnaire is presented in online supplementary appendix A.

### Modification of questionnaire by expert panel to obtain satisfactory content validity

A panel of experts in the areas of CVD, health psychology, public health, psychometrics and questionnaire development and medicine were asked to evaluate each item and the total 85-item questionnaire for content validity in February 2014. Experts assessed content validity of the questionnaire by examining whether the items were representative of the content they were intended to measure.[29] Items were examined for representatives of the scale domain, appropriateness and relevance. The content validity index (CVI), a widely used technique in scale development determined item and questionnaire clarity, homogeneity and relevance on a 4-point Likert scale (ranging from 1=*an irrelevant item* to 4=*an extremely relevant item*).[30 31] A CVI of ≥0.80 is recommended.[32 33] Experts were asked the following questions: '*Do these items belong together in the subscale?*' and '*Does each item belong in the set?*' For ratings of content validity, experts were asked whether the subscale definition and label fitted the set of items presented; whether each item belonged with the label and definition and whether each item was unique in its contribution to the subscale.

### Modification of questionnaire by patient focus group to obtain satisfactory face validity

Researchers facilitated a patient focus group to assess the face validity of the 69-item questionnaire resulting from the expert review. Face validity is assessed by end users deciding whether the questionnaire appears to measure what the researchers who developed it claim.[33] A convenience sample of six individuals was recruited on 2 March 2014 from the County Durham and Darlington NHS Foundation Trust. Eligibility criteria were being aged 40–74 years and being free of known vascular disease. The focus group consisted of six white females between 50 and 64 years of age. Most participants had postgraduate education. These individuals worked as clerical workers, nurses and health improvement staff. They were not involved in the delivery of the NHS Health Check programme. Participants were asked to complete the 69-item questionnaire as well as to provide feedback on whether the items correctly measured the intended scales, appropriately stated the intent of the questionnaire and matched the individual's situations.[32 33] In addition, participants were asked to respond to questions about clarity, content, appropriateness, format, biases of questions and presentation of information. The resulting 65-item questionnaire is presented in online supplementary appendix B.

### Phase II of questionnaire development
#### Modification of questionnaire to have satisfactory reliability

A 65-item questionnaire was administered to 110 NHS Health Check attendees immediately after their consultation between 21 May 2014 and 28 July 2014 in a cross sectional study in England. The aim was to determine the content, the scale structure and the reliability of the resulting questionnaire.

#### Study recruitment

Eligibility criteria were completion of an NHS Health Check, being aged 40–74 years and free of known vascular disease. Of 110 study participants, 15 individuals were recruited by 2 nurses from a London general practice and 95 individuals by 13 community outreach providers from local community venues in Durham. These providers collected clinical risk factor data, informed study participants about their CVD risk, took informed study consent and distributed the 65-item questionnaire to be self-completed by NHS Health Check attendees following their consultation. Unlike general practice staff who operated only during business hours, community outreach providers worked on evenings and weekends as well as during regular business hours in community venues more accessible to the general public.

#### Data analysis

To select appropriate items to constitute a scale, individual items were assessed during item analysis, item facility and item discrimination.[34] To determine the factorial structure of the questionnaire and which items together constituted particular scales, an exploratory factor analysis (EFA)—a widely used technique in scale development was performed.[30 35] The reliability of factors constituting particular scales was assessed using Cronbach's α coefficient.[36 37] Reliability refers to consistency, reproducibility and agreement of a scale.[38]

To improve the quality of a scale and increase its reliability, individual items were assessed. Items with reverse scoring were recoded to conform to the conceptual direction of the scales.[37] Each individual item was then examined for distortions in the pattern of responding known as skew and kurtosis.[33] Item facility examined whether items were answered in the same way by everyone by checking whether the facility index approached extreme scores or had a low SD.[34] Items were assessed in discriminating between participants' responses to the questionnaire's scales (Knowledge, Perceived CVD Risk, CVD Health

**Table 1** Sample item wording modifications obtained through an expert panel

| Original item(s) | Expert comments | Final item |
|---|---|---|
| The most important cause of heart attack and stroke is stress. | Revise to 'one of the most important…' Substitute the word 'important' with 'main'. | One of the main causes of heart attack and stroke is stress. |
| I have a high chance of getting a heart attack or stroke because of my past behaviours. | Add 'and/or present behaviours'. | I have a high chance of getting a heart attack or stroke because of my past and/or present behaviours. |
| Increasing my exercise will decrease my chances of having a heart attack or stroke. | Define amount of exercise. | Increasing my exercise to at least 30 min a day will decrease my chances of having a heart attack or stroke. |
| Eating a healthy diet will decrease my chance of having a heart attack or stroke. | Define a healthy diet. | Eating at least five portions of fruit and vegetables a day will decrease my chances of having a heart attack or stroke. |
| When I exercise I am doing something good for myself. | Define exercise consistently. Make the statement more specific about the heart. | When I exercise for 30 min a day I am doing something good for the health of my heart. |
| How confident are you that you know or can…? questions answered using a 5-point Likert scale: 'not at all confident, somewhat confident, moderately confident, very confident, completely confident'. | Use a 4-point Likert to maintain consistency. | Five-point Likert scale changed to a 4-point Likert scale: 'not at all confident, somewhat confident, very confident, completely confident'. |
| How confident are you that you know how or can stop smoking if you want to? | Instead of saying '…that you know or can' say 'that you know how to or can…' Add in parentheses 'if you smoke'. | How confident are you that you know how to or can stop smoking if you want to (if you smoke)? |
| I want to cut down on alcohol. I intend to cut down on alcohol in the next 2 months. | Conceptual overlap between want to and intend to. Add in parentheses 'if you drink alcohol'. | I intend or want to cut down on alcohol (if you drink alcohol). |

Beliefs, Intentions/Readiness to Change and Self-Efficacy). Discrimination was measured by item-total correlation with item correlating below 0.2 or any negative correlations resulting in deletion of items. In addition, discrimination was measured by the interitem correlation within each scale resulting in deletion of items correlating with other items ≥0.60.[17 34]

A Kaiser-Meyer-Olkin (KMO) measure of sampling adequacy and a Bartlett's test of sphericity were assessed to ensure that items were appropriate for EFA.[39] Next, EFA was performed to define the scales of the questionnaire which share a similar underlying construct. Parallel analysis was used to determine the optimum number of factors to be extracted using principal components analysis (PCA) with a Varimax rotation.[34 39 40] PCA is a data reduction technique used to explain correlations among sets of items or variables as a few conceptually meaningful factors.[30] Compared with other available methods, parallel analysis using PCA was shown to be the best method of extracting factors and is appropriate when applied to data conforming to the formal factor analytic model.[39 40] Iterations of EFA were carried out to identify core constituent items in each factor. Cross-loading items or items with loading≤0.50 were removed at each iteration.[39] Internal consistency reliability of resulting factors was assessed using Cronbach's α coefficients with α ≥0.70

indicating good reliability.[32 36 37] Associations between resulting factors and predicted CVD risk were examined using Spearman's rank correlation coefficient.

## RESULTS
### Construction of a draft questionnaire by review of relevant literature
We developed an 85-item questionnaire based on the theoretical framework, NHS guidelines and other validated questionnaires relating to heart disease.[17–19] The 85-item questionnaire had 8 subscales measuring *Knowledge of CVD Risk and Prevention* (18 items), *Perceived Risk and Vulnerability of CVD* (20 items), *Perceived Susceptibility* (5 items), *Perceived Severity* (5 items), *Perceived Benefits* (6 items), *Perceived Barriers* (7 items), *Self-Efficacy* (6 items) and *Intention to Change Behaviour* (18 items). *Knowledge of CVD Risk and Prevention* subscale items were measured using the following categories: *true, false* and *do not know*. *Self-Efficacy* subscale items were measured using 5-point Likert scale ranging from *1=not at all confident* to *5=completely confident*. *Perceived Severity, Perceived Benefits, Perceived Barriers* and *Intention to Change Behaviour* subscale items were measured using a 4-point Likert scale ranging from *1=strongly disagree* to *4=strongly agree*. The reading level of the questionnaire was at year 7.

**Table 2** Sample item wording modifications and additions through the patient focus group

| Original item | Participant comments | Final item |
|---|---|---|
| Moderate physical activity of 150 min a week will reduce your chances of developing a heart or stroke. | 2.5 hours a week is better than 150 min. | Moderate physical activity of 2.5 hours a week will reduce your chances of developing a heart or stroke. |
| Drinking alcohol has nothing to do with reducing the risk of heart attack or stroke. | Question is negatively stated. | Drinking high levels of alcohol can increase your cholesterol and triglyceride levels. |
| Missing question | Need to include family history of disease to account for genetic predisposition. | A family history of hypertension is not a risk factor for high blood pressure. |
| Missing question | Benefits of not smoking? | If I stopped smoking it will reduce my chances of having a heart attack or stroke. |
| Increasing my exercise for 30 min a day will decrease my chances of having a heart attack or stroke. | Two and a half hours a week is better than 30 min a day. | Increasing my exercise to at least 2 ½ hours a week will decrease my chances of having a heart attack or stroke. |
| I have reduced or stopped smoking (if you smoke). 'strongly disagree, disagree, agree, and strongly agree'. | Remove (if you smoke). Add a 'not applicable' box. | I have reduced or stopped smoking. 'strongly disagree, disagree, agree, and strongly agree, not applicable'. |
| How confident are you that you know how to or can consume recommended levels of alcohol (if you drink alcohol)? 'not at all confident, somewhat confident, very confident and completely confident'. | Remove (if you drink alcohol). Add a 'not applicable' box. | How confident are you that you know how to or can drink within the recommended levels of alcohol? 'not at all confident, somewhat confident, very confident and completely confident, not applicable'. |

## Modification of questionnaire by expert panel to obtain satisfactory content validity

The expert panel concluded that out of the 85 items, 69 met the CVI≥0.80 criterion and were retained. In addition, the wording of a number of questions was revised to improve clarity. Diet and exercise were defined more precisely using frequency and duration. Response options of Self-Efficacy items were changed from a 5-point Likert scale to a 4-point Likert scale for consistency with the rest of the questionnaire. Questions pertaining to smoking and drinking were rephrased to apply to smokers and drinkers (see table 1).

## Modification of questionnaire by patient focus group to obtain satisfactory face validity

As a result of the focus group review of the 69-item questionnaire, 6 items were removed, 2 items were added and a number of items were modified leaving a final total of 65 items with satisfactory face validity. A not applicable category was added to 50 items while the response categories to Knowledge subscale items remained unchanged. Exercise was redefined in 8 items from 150 min a week and 30 min a day to 2.5 hours a week. A negatively framed question was reframed positively (see table 2).

## Modification of questionnaire to have satisfactory reliability

The 65-item questionnaire that resulted from content and face validity assessments, was administered to 110 NHS Health Check attendees immediately after their NHS Health Check consultation. Most study participants were white (84.5%), younger than 60 (77.3%)

and had at least one or more CVD risk factors. Using the Index of Multiple Deprivation, a relative measure of deprivation across seven distinct domains including income, employment, health and disability, education skills and training, barriers to housing and services, living environment and crime,[41] people in the two most deprived fifths were 40.9% of the study population (see online supplementary appendix C) for study population characteristics. The responses to the questionnaire were analysed as individual items during item analysis, item facility and item discrimination. In addition, the scale structure and reliability of resulting scales were assessed.

No items were removed during item analysis and item facility. During item discrimination assessment using item-total correlation, seven items in the Knowledge scale, four items in Perceived CVD Risk, three items in CVD Health Benefits, three items in Intention and or Readiness to Change were deleted due to item-total correlations falling below 0.2.[33] During item discrimination assessment using interitem correlation, two items in Perceived CVD Risk and three items in Intentions/Readiness to Change were removed as these items correlated >0.6 with other items.[33] Although there were two items that correlated above 0.6 in CVD Risk Reduction Self-Efficacy, these remained in the questionnaire as the items were qualitatively different: *Stop smoking if you want to* and *Control the risks of having a heart attack or stroke*. In total, 22 items were removed during item discrimination analysis, leaving 43 items which had good item facility and discrimination.

**Table 3** Factor structure of the ABCD Risk Questionnaire

| | Components | | |
| --- | --- | --- | --- |
| | Factor 1 Perceived risk of heart attack/ stroke | Factor 2 Perceived benefits and intentions to change | Factor 3 Healthy eating intentions |
| It is likely that I will suffer from a heart attack or stroke in the future. | 0.844 | | |
| It is likely that I will have a heart attack or stroke some time during my life. | 0.816 | | |
| I feel I will suffer from a heart attack or stroke sometime during my life. | 0.809 | | |
| There is a good chance I will experience a heart attack or stroke in the next 10 years. | 0.752 | | |
| I am not worried that I might have a heart attack or stroke. | 0.705 | | |
| My chances of suffering from a heart attack or stroke in the next 10 years are great. | 0.687 | | |
| It is likely I will have a heart attack or stroke because of my past and/or present behaviours. | 0.639 | | |
| I am concerned about the likelihood of having a heart attack or stroke in the near future. | 0.575 | | |
| I am thinking about exercising at least 2 ½ hours a week. | | 0.826 | |
| I intend or want to exercise at least 2 ½ hours a week. | | 0.792 | |
| When I exercise for at least 2½ hours a week I am doing something good for the health of my heart. | | 0.735 | |
| I am confident that I can maintain a healthy weight by exercising at least 2½ hours a week within the next 2 months. | | 0.658 | |
| I am not thinking about exercising for 2 ½ hours a week. | | 0.656 | |
| When I eat at least five portions of fruit and vegetables a day I am doing something good for the health of my heart. | | 0.642 | |
| Increasing my exercise to at least 2½ hours a week will decrease my chances of having a heart attack or stroke. | | 0.557 | |
| I am confident that I can eat at least five portions of fruit and vegetables per day within the next 2 months. | | | 0.830 |
| I am thinking about eating at least five portions of fruit and vegetables a day. | | | 0.772 |
| I am not thinking about eating at least five portions of fruit and vegetables a day. | | | 0.731 |

Factor loadings and commonalities are reported following an EFA using principal component analysis with Varimax rotation.

Of the 43 remaining items, 8 items of the 'Knowledge' scale with *'true'* or *'false'* scoring could not be entered into EFA. Of the 35 items scored on a 4-point Likert scale, four items pertaining to smoking were deleted as they had a high proportion of missing responses (69%–80%). The resulting 31 items had a KMO measure of sampling adequacy of 0.32 and a significant Bartlett's test of sphericity (1020.50, p<0.001), indicating that these data were appropriate for EFA.[39] After 12 iterations of EFA, 20 items loaded above 0.50 on the factors and there were no cross-loadings indicating good factor structure (see table 3). Internal consistency reliability of factor structure was measured using Cronbach's α. Factor 1 (eight items): (Perceived Risk of Heart Attack/Stroke) had α=0.85.

Factor 2 (seven items): (Perceived Benefits and Intentions to Change) had α=0.82. Factor 3 (three items): (Healthy Eating Intentions) had α=0.56. Factor 4 (two items): (Intentions towards Alcohol) had α=−0.16. Although Healthy Eating Intentions α=0.56 is below the minimum threshold (0.70) for reliability, this is acceptable for a three-item scale.[34] The intention towards alcohol factor had two items with such low reliability (α=−0.16) that they could not be considered a separate factor and were removed. A 13th EFA iteration confirmed the factor loadings and reliabilities reported above. Hence, the parallel analysis indicated that three factors should be retained.[39] The three-factor model accounted for 57.61% of the total explained variance.

The EFA revealed three scales: Perceived Risk of Heart Attack/Stroke, Perceived Benefits and Intentions to Change and Healthy Eating Intentions. A fourth scale assessing Knowledge of CVD Risk and Prevention (not entered into EFA) was added back to the questionnaire following EFA (see figure 2). Hence, the resulting questionnaire included 26 items grouped into four scales: Knowledge of CVD Risk and Prevention (eight items), Perceived Risk of Heart Attack/Stroke (seven items), Perceived Benefits and Intention to Change Behaviour (seven items) and Healthy Eating Intentions (three items). In the resulting 26-item questionnaire, two items were changed from questions "How confident are you that you know how to or can…" to statements of agreement "I am confident that I can" so as to be answered using the same Likert scale. The time to complete this questionnaire is between 10 and 15 min. The ABCD Risk Questionnaire with a scoring guide for each scale is reported in online supplementary appendix D. Using Spearman's rho, there was a positive and significant relationship between perceived and predicted CVD risk (see online supplementary appendix E).

## DISCUSSION

To the best of our knowledge, this is the first study that describes the development of a short, validated questionnaire with satisfactory content and face validity and reliability examining CVD risk awareness among the NHS Health Check attendees. The ABCD Risk Questionnaire may be used for evaluating the accuracy of perceived CVD risk, general knowledge of CVD and intention to change behaviour in regard to diet and exercise among NHS Health Check attendees. Agreement between perceived and predicted CVD risk suggests that the tool performs well in assessing perceived CVD risk. As the questionnaire was developed using both an expert panel and a patient focus group, it ought to be relatively easy to understand for both patients and clinicians. If NHS Health Check recommendations change over time, it may need to be updated.

Critics of the NHS Health Check programme point to the lack of its evidence base.[42 43] The majority of evaluations focused on coverage and uptake, statin prescribing, new diagnoses and CVD risk factor reduction.[44–49] As there was no instrument measuring CVD risk awareness, no studies examined the patients' understanding of CVD risk among NHS Health Check attendees. CVD risk presentation was shown to increase the accuracy of perceived risk by 10%. When risk information is repeated, this leads to small but significant reductions in predicted CVD risk.[16] A national study showed modest reductions in 10-year predicted CVD risk among NHS Health Check attendees in the first 4 years.[48] A limitation of using predicted 10-year risk of CVD is the underestimation of CVD risk among women and younger people.[35] More research is needed to establish whether the programme improves NHS Health Check attendees' awareness of CVD risk and whether the programme has an impact on predicted lifetime CVD risk.

The ABCD Risk Questionnaire was developed on a non-risk stratified population after their initial NHS Health Check consultation as the NHS Health Check programme is administered to all eligible people free of vascular disease diagnosis irrespective of their level of CVD risk. The questionnaire does not encompass all aspects of CVD risk observed in the general population. Questions on smoking and drinking were progressively eliminated as they did not apply to most study participants. As questions on diet and exercise pertained to all people regardless of their level of CVD risk, such questions that reliably distinguished between study participants were selected for inclusion. Although fruit and vegetable intake is only one aspect of diet in the EatWell Guide recommended for use in NHS Health Check,[50] it is the only assessment of diet recorded during the NHS Health Check. The resulting questionnaire contains questions based on data collected during NHS Health Check to enable subsequent programme evaluation.[51] Future studies examining populations at increased CVD risk can look into incorporating smoking and alcohol into the ABCD Risk Questionnaire to learn about these individuals' preconceptions and attendance of follow-up care.

Judging by the number of items reduced in various stages of development, the ABCD Risk Questionnaire was largely shaped by analysis of data from 110 NHS Health Check attendees completing the 65-item questionnaire. This study population was representative of the population that took up the NHS Health Check programme between 2009 and 2014 in terms of sociodemographics including the proportion of men (46.4%), ethnic minorities (5.4%), individuals from the most deprived two-fifths (40.9%), and clinical risk factors including mean total cholesterol (5.42 (95% CI 5.19 to 5.64)), body mass index (27.24 95% CI 26.17 to 28.31), smokers (18.2%) and those at high CVD risk (4.5%).[44] As higher levels of deprivation are partly due to having less education,[41] questionnaire development was not limited to people with higher education. Compared with the national evaluation, similar levels of high CVD risk were observed despite the fact that the study population contained more younger people aged 40–59 years (77.3%).[44] The recruitment of hard-to-reach groups including younger people, socioeconomically deprived individuals and ethnic minorities by community outreach providers in community venues outside of conventional working hours is consistent with prior literature.[22 52–54]

A possible limitation to face validity is that the patient focus group evaluating the 69-item questionnaire was not representative of the target population. Whereas the NHS Health Check programme is administered to both men and women and members of ethnic minorities, the focus group consisted only of white women. Furthermore, as these women had postgraduate education and worked in a health-related field, they may have had higher health literacy than the general population eligible for the NHS

Health Check programme. Clarity, appropriateness, biases and presentation of information may have been differentially assessed by people with different levels of health literacy. A community-based recruitment method aiming to recruit some of the hard-to-reach groups may have been more effective in getting a more representative patient focus group.

Additional studies should be conducted with larger samples to confirm the reliability and validity of the questionnaire. It would be useful to replicate the factor analytic process on an independent, larger sample to confirm the generalisability of these findings.[37]

## CONCLUSIONS

The ABCD Risk Questionnaire showed evidence of satisfactory reliability and validity, is brief and easy to use. By capturing patients' views on CVD risk awareness during an NHS Health Check consultation, the questionnaire can be used to assess patients' understanding of CVD risk. Clinicians administering the questionnaire to patients may help to establish whether the programme is effective in empowering patients to make informed lifestyle choices about their health.

**Author affiliations**
[1]Department of Primary Care and Public Health, Imperial College London, London, UK
[2]Department of Psychology, City University London, London, UK
[3]Health Diagnostics Ltd, Chester, UK
[4]Department of Applied Health Research, University College London, London, UK
[5]Institute of Global Health Innovation, Imperial College London, London, UK

**Contributors** MW, AM, MS and HW designed the study, JE supplied the data. JJN designed the validation instrument, LZ performed the psychometric analysis. JE, AK, MH and AM reviewed the validation instrument's face and content validity. All authors discussed data analyses and interpreted the results. MW wrote the first draft of the manuscript. All authors critically revised and approved the final manuscript. MW had full access to all the data used in the study and takes responsibility for the integrity of the data and the accuracy of the data analysis. MW is the guarantor.

**Funding** This work was supported by the National Institute for Health Research (NIHR) Diagnostics Evidence Co-operatives (DEC)/Collaboration for Leadership in Applied Health Research and Care (CLAHRC) grant.

**Competing interests** None declared.

**Ethics approval** NRES Committee London—City and East.

**Provenance and peer review** Not commissioned; externally peer reviewed.

**Data sharing statement** No additional data available.

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
