## [Reviewer comments · BMJ Open]

ARTICLE DETAILS

TITLE (PROVISIONAL)	Development of a questionnaire to evaluate patients' awareness of cardiovascular disease risk in England's National Health Service Health Check preventive cardiovascular programme
AUTHORS	Woringer, Maria; Nielsen, Jessica; Zibarras, Lara; Evason, Julie; Kassianos, Angelos; Harris, Matthew; Majeed, Azeem; Soljak, Michael

VERSION 1 – REVIEW

REVIEWER	Prof Tom Cochrane University of Canberra, Australia
REVIEW RETURNED	01-Oct-2016

GENERAL COMMENTS	The authors report their work on an important gap in CVD prevention research – a validated instrument to measure an individual's knowledge of CVD risk, perceptions of risk and intention to change behaviour. They, quite rightly, point out that such a tool would be helpful in the evaluation of the effectiveness of a CVD risk assessment and management programme, such as the NHS Health Check programme. The first phase of development of the questionnaire was guided by a literature review, an expert panel and a patient focus group. This resulted in an 85 item draft version of the questionnaire, which was reduced to a 65 item instrument for use in phase two of the project. Phase two used the 65 item version with a participant sample of 110 NHS Health Check attendees, which resulted, following further Exploratory Factor Analysis (to determine item loading and the key constructs to be measured) and Cronbach's alpha (to determine reliability), in a 26 item final version of the questionnaire. The authors have made a laudable effort to develop a questionnaire with satisfactory face and content validity and reliability. However, I detected issues with some items in all three versions of the questionnaire that I believe the authors should consider. 85 item first draft questionnaire: • Item 10 appears to need the word 'attack' after 'heart'• Item 15 is poorly worded – patients may think they just need to breathe rapidly for a sustained period of time. It would be better to say 'The healthiest exercise for the heart is exercise that raises the heart rate and breathing rate for a sustained period of time.' (could even be more specific here with a given time frame reference)• Items 17, 66, 83, 84, 85 may have ambiguous alternatives. For example, many people believe (the majority of current population norms are used as a guide) that being overweight is good for CVD
---

	health whereas most would agree that obesity is not.  • Items 51, 59, 64, 74, 77 only apply to those who smoke • Items 54, 65, 75, 79, 81 only applies to those who drink alcohol 65 Item phase two draft questionnaire:  • Item 10 is the same poorly worded item referred to above • Item 12 has potentially ambiguous alternatives • Items 46 and 61-65 • Items 47 and 57-60 only applies to drinkers 26 item final version:  • Knowledge and Perceived risk (Items 1-16) seem satisfactory • Perceived benefits and Intentions to change sections may have been better as separate factors. It is possible to perceive the benefits but have no intention to change or vice versa. Also, some of the wording may cause issues with completion. For example, item 19 only applies to those who exercise at least 2½ hours per week, item 22 only to those eating 5 portions of fruit and vegetables a day and item 23 only to those increasing their exercise to at least 2½ hours a week. • Items 25 and 26 in the Healthy eating intentions section are mutually exclusive • This questionnaire evaluates only a very limited subset of the risk factors or behaviours pertinent to circulatory disease in the UK. Overall, then, I believe the questionnaire in its present form does have some merit but falls short on evaluation of potential influence of the NHS Health Check on the specific modifiable behaviours or risk factors that are known to contribute most to cardiovascular and circulatory disease risk in the UK. The most recent Global Burden of Disease Report Card for the UK indicates that these are: dietary (10%), high blood pressure (8.8%), high BMI (4.7%), smoking (3.6%), physical inactivity (3.4%) i.e. potentially 30.5% of the burden of circulatory disease may be amenable to prevention through modification of just these five factors. The sole focus in the Healthy eating intentions section on consumption of five portions of fruit and vegetables is highly unlikely to capture adequately the potential influence of the NHS Health Check on dietary behaviours. Furthermore, there is no reference to the related issue of adherence to or tolerance of medication, which is known to be a challenge for many patients. There were a number of minor points that the authors may wish to address: p6, l29/30 Criteria is the plural of criterion p7, l36/7 Explain abbreviations CVD probably OK but HBM may not be so well known p8, l9/10 Criteria plural again p13, l28/9 Insert word 'be' after 'not' p15, l38/9 'commonalities' should read 'communalities' p17, l20/1 To substantiate 'brief' questionnaire in Conclusions, Results section should provide some indication of mean and spread of completion times
--	---

REVIEWER	Chris Clark Clinical Senior Lecturer in General Practice Primary Care Research Group, Smeall Building, St Luke's Campus, Exeter, UK, EX1 2LU
REVIEW RETURNED	26-Oct-2016

GENERAL COMMENTS

This paper describes the development and validation of a questionnaire to assess CVD risk awareness resulting from participation in a NHS Health Check. The authors state their aim to be "to develop a questionnaire...to assess the effectiveness of the NHS Health Check programme in raising patient's awareness of CVD risk" (page 5 lines 10-12). Unfortunately they have not done this. To demonstrate raised awareness would surely require before and after comparison or similar techniques. This is purely a questionnaire development study and the aims should be restricted to that. This also applies to the implication that the questionnaire itself can provide evidence of effectiveness (page 4 last lines).

The questionnaire was designed initially through literature searches. An expert panel assessed content validity then a patient focus group assessed face validity. They are described as a patient focus group, yet are in fact six white females from an NHS Foundation Trust (employed or otherwise is not stated). They were all aged 50-64, whereas the Health Check targets 40 to 74 year olds, mainly held postgraduate qualifications and worked in a "health related field" (page 6 paragraph 3). I believe that this is an important weakness that needs thorough discussion and justification because: 1) there are no men, and we know men are harder to reach in the NHS HC programme; 2) we do not know if the sample have undergone an NHS HC; 3) what sort of health workers are they - could they be nurses involved in delivering HCs or other cardiovascular risk assessment? 4) they all have postgraduate education and we also know that those with lower socio-economic status, associated with educational attainment, are also less likely to attend a HNS HC. One is left with the impression that face validity has been assessed by a selected small group motivated to attend (or perhaps even deliver) NHS HCs, and be predisposed to greater personal awareness of CV risk anyway. For me this is an important weakness of design.

Study population: The questionnaire was administered to 110 participants in NHS Health checks - a table describing their demographics - age range, gender, ethnicity, and health check findings is presented as appendix C but should be included in the body of the paper to better describe the population, means and variance or range would be informative. It seems that the nurses undertaking the health checks also administered the questionnaire. This means, I think, that they must have been aware of the individuals' CV risk and is a potential source of bias that should be clarified and discussed.

The data analysis section is detailed but not my area of expertise hence my recommendation of a statistical review from an expert in the area.

The first paragraph of the discussion tells us how the ABCD questionnaire might be assessed against recorded clinical risk factors. It seems an important omission that the authors have not done this, since to do so would have addressed the unfulfilled aim referred to above. They appear to have a dataset of 110 participants with full recording of risk markers and QRisk2 score. It should be possible to examine correlations of these data with the perceptions of risk scores as they suggest. Whilst the sample size may not be adequate for robust conclusions even this pilot analysis would complete the paper.

REVIEWER	Gordon Prescott University of Aberdeen, UK
REVIEW RETURNED	19-Apr-2017

GENERAL COMMENTS	The abstract is not very clear. The results appear in the methods section of the abstract. In the abstract and elsewhere it is difficult to keep track of the number of scales being considered at that point in the study. The strengths and limitations of the study listed by the abstract are mixed together so it is not clear which are strengths and which are weaknesses. The focus group section appears to be the weakest element of this study. The focus group has only white, female participants with the majority having postgraduate education and employment in healthcare. It was acknowledged that they were not to be representative of the target population. Why wasn't this addressed by recruiting a second focus group which was more diverse? The lack of any diversity might have influenced the responses on clarity, appropriateness, biases and presentation. People from different backgrounds, with lower levels of education or with greater levels of cardiovascular risk might have had very different views. This is acknowledged in the discussion, but is very much downplayed because the participants completing the revised questionnaire were more diverse and did include "a large proportion of deprived individuals". Perhaps the potential consequences of the weakness of this focus group stage need greater consideration. The method of selection of 110 Health Check attendees who make up the study population is not entirely clear or justified. There does not appear to be any formal method of selection or any dates given for a period of recruitment of eligible people. Why do only 15 come from a London General Practice and 95 from local community venues in Durham? This group of 110 is stated to be "representative" of the target population, but little evidence is presented for this apart from the percentages with high CVD risk and also smoking among the participants. There are too few people in the group of 15 for there to have been any investigation of whether there were any differences between these from a GP practice and the rest from community venues. The recruitment method(s) need more coverage if the inferences from the quantitative section of the paper are to be accepted. In general, the paper needs to be a little more carefully justified. Typos and formatting: p9 Table 1 typo "strike" should be "stroke". P15 line 5 typo: "suggests" should be "suggest". Table 3 has two statements which appear out of place because they do not relate to the person's beliefs, perception of risk or intentions: "Maintain a healthy weight by exercising at least 2 1/2 hours a week within the next two months" and "Eat at least five portions of fruit and vegetables per day within the next two months". These differences in the types of statement may be correct, but they look odd. In Table 3 Page 14, line 24 there is a typo of an extra "l". The first flow diagram has one box on the right which is too small
---

	and has truncated the text inside. Appendix A has some formatting issues. The questions on Self Efficacy do not have question marks. Appendix B has some formatting issues. In the exercise section the spacing around 2.5 as a fraction is not consistent. Appendix C has a problem in the last few lines as it is not clear if exactly 20% is high risk or medium risk as 20% could be allocated to both of these groups.
--	---

VERSION 1 – AUTHOR RESPONSE

Reviewer 1 Comment to Author:

The authors have made a laudable effort to develop a questionnaire with satisfactory face and content validity and reliability. However, I detected issues with some items in all three versions of the questionnaire that I believe the authors should consider.

85 item first draft questionnaire:

- Item 10 appears to need the word ‘attack’ after ‘heart’
- Item 15 is poorly worded – patients may think they just need to breathe rapidly for a sustained period of time. It would be better to say ‘The healthiest exercise for the heart is exercise that raises the heart rate and breathing rate for a sustained period of time.’ (could even be more specific here with a given time frame reference)
- Items 17, 66, 83, 84, 85 may have ambiguous alternatives. For example, many people believe (the majority if current population norms are used as a guide) that being overweight is good for CVD health whereas most would agree that obesity is not.
- Items 51, 59, 64, 74, 77 only apply to those who smoke
- Items 54, 65, 75, 79, 81 only applies to those who drink alcohol

65 Item phase two draft questionnaire:

- Item 10 is the same poorly worded item referred to above
- Item 12 has potentially ambiguous alternatives
- Items 46 and 61-65
- Items 47 and 57-60 only applies to drinkers

Author Response:

I inserted the word ‘attack’ after ‘heart’ in the 85 item questionnaire as this was an obvious typo. As for the rest of the items in the 85 and the 65 item questionnaire, it is not possible to change the wording retrospectively. Development of the initial 85 item questionnaire was guided by literature review of other validated questionnaires which resulted in the inclusion of 65 items.

In addition, 23 new items were generated. These new items were grounded in psychology theory, pertained to data collected during an NHS Health Check and were based on behaviour specific recommendations of the programme (p 7 lines 15-26). The development of the questionnaire was subsequently a reductive process guided by an expert panel that assessed the content validity of the questionnaire as well as by a patient focus group that assessed its face validity.

Reviewer 1 Comment to Author:

26 item final version:

- Knowledge and Perceived risk (Items 1-16) seem satisfactory
- Perceived benefits and Intentions to change sections may have been better as separate factors. It is possible to perceive the benefits but have no intention to change or vice versa. Also, some of the

wording may cause issues with completion. For example, item 19 only applies to those who exercise at least 2½ hours per week, item 22 only to those eating 5 portions of fruit and vegetables a day and item 23 only to those increasing their exercise to at least 2½ hours a week.

- Items 25 and 26 in the Healthy eating intentions section are mutually exclusive

Author Response:

As to the 26 item final version, the factor structure of the scale were derived from the analysis of data. It is not possible to change the content of the factors even if as you rightly point out, perceived benefits and intentions to change may have been better as separate factors. The questions on diet and exercise only apply to those who consume 5 portions of fruit and vegetables a day and those exercising to at least 2 ½ hours per week. However the responses do include a Not Applicable option. And those to whom such questions do not apply, can select such an option. Items 25 and 26 are not mutually exclusive as item 26 is reverse coded. Hence if scored correctly and summed together, items 25 and 26 would help to establish healthy eating intentions.

Reviewer 1 Comment to Author:

This questionnaire evaluates only a very limited subset of the risk factors or behaviours pertinent to circulatory disease in the UK. Overall, then, I believe the questionnaire in its present form does have some merit but falls short on evaluation of potential influence of the NHS Health Check on the specific modifiable behaviours or risk factors that are known to contribute most to cardiovascular and circulatory disease risk in the UK. The most recent Global Burden of Disease Report Card for the UK indicates that these are: dietary (10%), high blood pressure (8.8%), high BMI (4.7%), smoking (3.6%), physical inactivity (3.4%) i.e. potentially 30.5% of the burden of circulatory disease may be amenable to prevention through modification of just these five factors. The sole focus in the Healthy eating intentions section on consumption of five portions of fruit and vegetables is highly unlikely to capture adequately the potential influence of the NHS Health Check on dietary behaviours. Furthermore, there is no reference to the related issue of adherence to or tolerance of medication, which is known to be a challenge for many patients.

Author Response:

In terms of the questionnaire falling short of evaluating specific modifiable behaviours or risk factors, the questionnaire is not inclusive of all risk factors pertaining to CVD. For instance the assessment of diet only consists of consumption of five fruits and vegetables a day. However this is the only measurable aspect of diet recorded during an NHS Health Check consultation. And as the questionnaire was made for evaluating the NHS Health Check programme, we chose to focus on quantifiable risk factors recorded during the NHS Health Check consultation. This is now made clear in the Discussion (p 16, lines 17-21). In terms of assessing other risk factors such as smoking and alcohol consumption, these were not included in the final questionnaire largely because such questions did not apply to the vast majority of respondents. Most of the 110 respondents to the 65 item questionnaire left questions pertaining to smoking unanswered.

Questions pertaining to drinking had too low of a reliability for questionnaire inclusion. Hence such questions could not be included in the final questionnaire. Future studies examining populations at increased CVD risk can look into these two scales (smoking and alcohol) and how they can be incorporated into the ABCD Risk Questionnaire. During the NHS Health Check consultation, patients that are identified to be at high risk are invited for further consultations where they may be prescribed cholesterol or blood pressure lowering medication. The questionnaire in development focused primarily on the first CVD risk assessment consultation and as such did not capture information related to medication prescribing and adherence among high CVD risk patients.

Reviewer 1 Comment to Author:

There were a number of minor points that the authors may wish to address:
p6, l29/30 Criteria is the plural of criterion

p7, l36/7 Explain abbreviations CVD probably OK but HBM may not be so well known
p8, l9/10 Criteria plural again
p13, l28/9 Insert word 'be' after 'not'
p15, l38/9 'commonalities' should read 'communalities'
p17, l20/1 To substantiate 'brief' questionnaire in Conclusions, Results section should provide some indication of mean and spread of completion times

Author Response:

The terms 'criterion,' 'criteria' and 'commonalities' were correctly used. Health Belief Model is now spelled out in the first instance. The term 'brief' refers to the final number of items 26 – rather than time completion.

Reviewer 2 Comment to Author:

This paper describes the development and validation of a questionnaire to assess CVD risk awareness resulting from participation in a NHS Health Check. The authors state their aim to be "to develop a questionnaire...to assess the effectiveness of the NHS Health Check programme in raising patient's awareness of CVD risk" (page 5 lines 10-12). Unfortunately they have not done this. To demonstrate raised awareness would surely require before and after comparison or similar techniques. This is purely a questionnaire development study and the aims should be restricted to that. This also applies to the implication that the questionnaire itself can provide evidence of effectiveness (page 4 last lines).

Author Response:

You are correct in highlighting the fact that this paper dealt with questionnaire development and not the evaluation of the NHS Health Check programme's effectiveness. I deleted mention of assessing 'effectiveness' from the study and instead refocused the paper on creating a novel tool for measuring CVD risk awareness.

Reviewer 2 Comment to Author:

The questionnaire was designed initially through literature searches. An expert panel assessed content validity then a patient focus group assessed face validity. They are described as a patient focus group, yet are in fact six white females from an NHS Foundation Trust (employed or otherwise is not stated). They were all aged 50-64, whereas the Health Check targets 40 to 74 year olds, mainly held postgraduate qualifications and worked in a "health related field" (page 6 paragraph 3). I believe that this is an important weakness that needs thorough discussion and justification because: 1) there are no men, and we know men are harder to reach in the NHS HC programme; 2) we do not know if the sample have undergone an NHS HC; 3) what sort of health workers are they - could they be nurses involved in delivering HCs or other cardiovascular risk assessment? 4) they all have postgraduate education and we also know that those with lower socio-economic status, associated with educational attainment, are also less likely to attend a HNS HC.

One is left with the impression that face validity has been assessed by a selected small group motivated to attend (or perhaps even deliver) NHS HCs, and be predisposed to greater personal awareness of CV risk anyway. For me this is an important weakness of design.

Author Response:

The lack of diversity in the focus group was given greater consideration in the Discussion section (p. 16, lines 37-45). Also the occupational roles of focus group participants were better described under "Phase I of Questionnaire Development, Modification of questionnaire by patient focus group to obtain satisfactory face validity" section of the paper (p.7, lines 51-53). The study participants did not deliver the NHS Health Check intervention. This is now clearly stated in the above section (p7 line 53, p. 8 line 7).

Reviewer 2 Comment to Author:

Study population: The questionnaire was administered to 110 participants in NHS Health checks - a table describing their demographics - age range, gender, ethnicity, and health check findings is presented as appendix C but should be included in the body of the paper to better describe the population, means and variance or range would be informative. It seems that the nurses undertaking the health checks also administered the questionnaire. This means, I think, that they must have been aware of the individuals' CV risk and is a potential source of bias that should be clarified and discussed.

Author Response:

The 110 participants of the NHS Health Check, as requested, are better described in terms of their clinical risk factors with means and 95% confidence intervals in Appendix C (p.31, lines 35-50). It was not possible to include this information as part of the main body as the paper is limited to the inclusion of 5 tables and figures and 3 tables and 2 figures were included. The healthcare professionals carrying out the NHS Health Check did not administer the questionnaire. These questionnaires were self-administered by patients and this point is made clearer in the "Study Recruitment" section (p7, lines 26-29).

Reviewer 2 Comment to Author:

The first paragraph of the discussion tells us how the ABCD questionnaire might be assessed against recorded clinical risk factors. It seems an important omission that the authors have not done this, since to do so would have addressed the unfulfilled aim referred to above. They appear to have a dataset of 110 participants with full recording of risk markers and QRisk2 score. It should be possible to examine correlations of these data with the perceptions of risk scores as they suggest. Whilst the sample size may not be adequate for robust conclusions even this pilot analysis would complete the paper.

Author Response:

As to the last point regarding assessing programme effectiveness by comparing predicted risk with clinical risk, I'm afraid that this is outside of the current scope of the paper. This point was addressed as per your earlier comment in the Discussion by omitting the aim of evaluating "programme effectiveness" altogether and refocusing the paper on questionnaire development. We still believe this to be a very important paper as it gives an instrument with which the NHS Health Check attendees' understanding of CVD risk may be evaluated.

Reviewer 3 Comment to Author:

The abstract is not very clear. The results appear in the methods section of the abstract. In the abstract and elsewhere it is difficult to keep track of the number of scales being considered at that point in the study.

The strengths and limitations of the study listed by the abstract are mixed together so it is not clear which are strengths and which are weaknesses.

Author Response:

Questionnaire development is a lengthy and complex process. The Methods section describes initial item development, subsequent expert panel, focus group review, and administration of the 65 item questionnaire to 110 NHS Health Check attendees. The Results are limited to the description of the resulting 26 item questionnaire. The formulation of the final 26 item questionnaire is of most interest to the study. Figures 1 and 2 were provided to help keep track of questionnaire development (by detailing the number of items in each questionnaire) throughout different stages of questionnaire development.

The strengths and limitations are as follows, though according to journal guidelines they were

grouped together under a single heading “Strengths and limitations of this study (p.4 lines 49-53, p.5 lines 7-14).”

Strengths of this study

- Questionnaire guided by literature review, expert panel, patient focus group & data analysis
- Largely developed among 110 individuals representative of the target population

Limitation of this study

- Face validity assessed via a patient focus group not representative of the target population

Reviewer 3 Comment to Author:

The focus group section appears to be the weakest element of this study. The focus group has only white, female participants with the majority having postgraduate education and employment in healthcare. It was acknowledged that they were not to be representative of the target population. Why wasn't this addressed by recruiting a second focus group which was more diverse? The lack of any diversity might have influenced the responses on clarity, appropriateness, biases and presentation. People from different backgrounds, with lower levels of education or with greater levels of cardiovascular risk might have had very different views. This is acknowledged in the discussion, but is very much downplayed because the participants completing the revised questionnaire were more diverse and did include “a large proportion of deprived individuals”. Perhaps the potential consequences of the weakness of this focus group stage need greater consideration.

Author Response:

The lack of diversity in the focus group was given greater consideration in the Discussion section. The Discussion section has been expanded to incorporate your comments (p. 16, lines 37-45).

Reviewer 3 Comment to Author:

The method of selection of 110 Health Check attendees who make up the study population is not entirely clear or justified. There does not appear to be any formal method of selection or any dates given for a period of recruitment of eligible people. Why do only 15 come from a London General Practice and 95 from local community venues in Durham? This group of 110 is stated to be “representative” of the target population, but little evidence is presented for this apart from the percentages with high CVD risk and also smoking among the participants. There are too few people in the group of 15 for there to have been any investigation of whether there were any differences between these from a GP practice and the rest from community venues. The recruitment method(s) need more coverage if the inferences from the quantitative section of the paper are to be accepted.

Author Response:

Exact dates are now given for recruitment both in the Abstract (p.4, line 27) and in the Methods of the paper (p.8, line 18) to be between May 21 and July 28, 2014. Regarding recruitment method, you are right in pointing out that this was not sufficiently addressed before, particularly as to why 15 people came from GP and 95 from community venues.

There were many more providers operating in the community (13) than in the GP (2) which may have been responsible for community outreach providers serving more Health Check attendees than nurses employed in general practice. Also community outreach providers worked outside normal business hours and in venues more accessible to the general public. There is now a “Study Recruitment” section of the paper describing this in more detail (p. 8, lines 24-31). The 110 NHS Health Check attendees' clinical risk factor profile and socio-demographics suggests that these individuals were representative of the target population. Your points regarding expansion of risk factors to be judged “representative” was taken into account and now there is a range of both socio-demographic and clinical risk factors presented as evidence to the above statement in the Discussion section (p. 16, lines 23-35) (and also in Appendix C (p.31, lines 35-50)).

Reviewer 3 Comment to Author:

Typos and formatting:

p9 Table 1 typo "strike" should be "stroke".

P15 line 5 typo: "suggests" should be "suggest".

Table 3 has two statements which appear out of place because they do not relate to the person's beliefs, perception of risk or intentions: "Maintain a healthy weight by exercising at least 2½ hours a week within the next two months" and "Eat at least five portions of fruit and vegetables per day within the next two months". These differences in the types of statement may be correct, but they look odd.

In Table 3 Page 14, line 24 there is a typo of an extra "I".

The first flow diagram has one box on the right which is too small and has truncated the text inside.

Appendix A has some formatting issues. The questions on Self Efficacy do not have question marks.

Appendix B has some formatting issues. In the exercise section the spacing around 2.5 as a fraction is not consistent.

Appendix C has a problem in the last few lines as it is not clear if exactly 20% is high risk or medium risk as 20% could be allocated to both of these groups.

Author Response:

The questions pertaining to intention in Table 3 are more clearly worded (p. 14, lines 24, 40). All typos and formatting issues have been corrected. The first flow diagram has been amended to include all the text (p.21, line 20). High risk and medium risk is more clearly delineated in Appendix C (line 33, p.31).

VERSION 2 – REVIEW

REVIEWER	Professor Tom Cochrane University of Canberra Australia
REVIEW RETURNED	16-May-2017

GENERAL COMMENTS	The reviewer also provided a marked copy with additional comments. Please contact the publisher for full details.
---

REVIEWER	Chris Clark University of Exeter Medical school UK
REVIEW RETURNED	12-May-2017

GENERAL COMMENTS	The authors appear to have addressed some of my comments. I am happy with the revisions to the text. I still believe that the table of
--

	characteristics of participants should be brought in to the main text from the appendices but that is clearly an editorial decision. The authors have neither examined correlations of their subjects cardiovascular risk scores with the perceptions of risk scores, nor addressed this as a weakness. I still think it would enhance this study enormously to show what correlation exists between the output of this risk perception questionnaire and actual validated QRisk2 score given in the health check. If there is no correlation then this is a very different project than if there is some measure of agreement.
--	--

REVIEWER	Gordon Prescott University of Aberdeen, Scotland
REVIEW RETURNED	08-May-2017

GENERAL COMMENTS	The recruitment of the 110 participants is much more clearly explained and justified by the authors in the methods and the abstract. Some summary statistics for the 110 appear in the text of the paper even if the full description can only be included in an appendix. The representativeness of the 110 Health Check attendees has been addressed much more fully. The point about the lack of representativeness of the focus group has also been addressed more comprehensively in the discussion. The authors have appropriately changed the paper to omit an assessment of effectiveness.
---

VERSION 2 – AUTHOR RESPONSE

Reviewer 1 Initial Comment to Author:

The authors have made a laudable effort to develop a questionnaire with satisfactory face and content validity and reliability. However, I detected issues with some items in all three versions of the questionnaire that I believe the authors should consider.

85 item first draft questionnaire:

- Item 10 appears to need the word 'attack' after 'heart'
- Item 15 is poorly worded – patients may think they just need to breathe rapidly for a sustained period of time. It would be better to say 'The healthiest exercise for the heart is exercise that raises the heart rate and breathing rate for a sustained period of time.' (could even be more specific here with a given time frame reference)
- Items 17, 66, 83, 84, 85 may have ambiguous alternatives. For example, many people believe (the majority if current population norms are used as a guide) that being overweight is good for CVD health whereas most would agree that obesity is not.
- Items 51, 59, 64, 74, 77 only apply to those who smoke
- Items 54, 65, 75, 79, 81 only applies to those who drink alcohol

65 Item phase two draft questionnaire:

- Item 10 is the same poorly worded item referred to above
- Item 12 has potentially ambiguous alternatives
- Items 46 and 61-65
- Items 47 and 57-60 only applies to drinkers

Initial Author Response:

I inserted the word 'attack' after 'heart' in the 85 item questionnaire as this was an obvious typo. As for the rest of the items in the 85 and the 65 item questionnaire, it is not possible to change the wording retrospectively. Development of the initial 85 item questionnaire was guided by literature review of other validated questionnaires which resulted in the inclusion of 65 items. In addition, 23 new items were generated. These new items were grounded in psychology theory, pertained to data collected during an NHS Health Check and were based on behaviour specific recommendations of the programme. The development of the questionnaire was subsequently a reductive process guided by an expert panel that assessed the content validity of the questionnaire as well as by a patient focus group that assessed its face validity.

Comment #1 to Author:

Whilst I accept that it is not possible to change the wording of the questions retrospectively, it should have been possible to get the wording right in the first place, irrespective of whether the questions are gleaned from published work or are grounded in psychology theory.

Both the 85 question and the 65 question versions that preceded the reductive phase of the study, in my view, contained several questions that were worded unsatisfactorily and I believe this limitation has carried through to the proposed final 26 item version of the questionnaire. Content validity may be partially satisfactory but the wording and scoring of individual questions is not

Author Response to Comment #1:

The reviewer makes a valid point regarding perhaps that some of the items noted in previous responses can be interpreted as unclear or could have been worded differently, and their points are taken. The author would like to reassure the reviewer that we have scrutinised the wording of the initial questionnaire with focus group members and expert panellists which incorporated discussion and responses pertaining to the clarity, appropriateness of information presented, the amount of information presented, biases, format of question and presentation of the information using wording directly from the NHS Health Check Personal Reports (documents given directly to NHS Health Check participants). Not once were the above issues raised by members of either group in the validation process. However, it should be noted that all items above identified by the reviewer are no longer presented in the brief 26-item questionnaire. We do recognise that no questionnaire can be worded perfectly for every respondent and that we aim to continue to review the psychometric properties of this questionnaire in future studies with larger sample sizes. As such, we now acknowledge these realities as limitations of our work.

Reviewer 1 Initial Comment to Author:

26 item final version:

- Knowledge and Perceived risk (Items 1-16) seem satisfactory
- Perceived benefits and Intentions to change sections may have been better as separate factors. It is possible to perceive the benefits but have no intention to change or vice versa. Also, some of the wording may cause issues with completion. For example, item 19 only applies to those who exercise at least 2½ hours per week, item 22 only to those eating 5 portions of fruit and vegetables a day and item 23 only to those increasing their exercise to at least 2½ hours a week.
- Items 25 and 26 in the Healthy eating intentions section are mutually exclusive

Initial Author Response:

As to the 26 item final version, the factor structure of the scale were derived from the analysis of data. It is not possible to change the content of the factors even if as you rightly point out, perceived benefits and intentions to change may have been better as separate factors. The questions on diet and exercise only apply to those who consume 5 portions of fruit and vegetables a day and those exercising to at least 2 ½ hours per week. However the responses do include a Not Applicable option. And those to whom such questions do not apply, can select such an option. Items 25 and 26 are not mutually exclusive as item 26 is reverse coded. Hence if scored correctly and summed together, items 25 and 26 would help to establish healthy eating intentions.

Reviewer 1 Comment #2

The factor structure was derived from a relatively small (110 patients), weakly representative sample using a set of 65 questions, many of which were poorly worded or ambiguous. The authors appear to accept this process with blind faith given that they argue that they see no need to alter the 26 item final version of their questionnaire.

I do not consider this a satisfactory response to my comments. Questions 17-26 contain questions that may have alternative interpretations or alternative divergent scoring options. Take Q23 for example:

Increasing my exercise to at least 2 hours a week will decrease my chances of having a heart attack or stroke.

A respondent who never exercises to this extent may interpret this question as not applying to me and therefore score 0 or N/A. On the other hand, they could equally decide to Strongly agree, score 1, because they agree that this would be good even though they have no intention of adopting this behaviour or Strongly disagree, score 4, because they don't believe that this action would be good for them at all or is not feasible for them under their circumstances. Thus, there are alternative scores dependent only on the respondent's interpretation of the question.

Including negative versions of the same basic question and then reverse coding may be acceptable as a validity check but will distort scoring since these items are effectively included twice. Scoring in any case will be complicated by the inclusion of a large number of N/A options that are difficult to interpret.

Author Response to Comment #2:

Whilst the authors agree with some of the points noted above, we would like to point out that the example provided by the reviewer is a prime example of rewording previous items based on the feedback received by the focus group members of the initial validation process. It was recommended by members of this group to reword the following from: "Increasing my exercise for 30 minutes a day will decrease my chances of having a heart attack or stroke," which is the wording used from the NHS Health Check Participant Information Leaflets to "Increasing my exercise to at least 2½ hours a week will decrease my chances of having a heart attack or stroke." Going forward will keep what the reviewer has mentioned in mind and consider in future versions of the questionnaire. It should also be noted that the respondents of this questionnaire also complete measures that check their actual risk factor indicators which this questionnaire are compared to for feedback purposes.

Reviewer 1 Comment to Author:

This questionnaire evaluates only a very limited subset of the risk factors or behaviours pertinent to circulatory disease in the UK. Overall, then, I believe the questionnaire in its present form does have some merit but falls short on evaluation of potential influence of the NHS Health Check on the specific modifiable behaviours or risk factors that are known to contribute most to cardiovascular and circulatory disease risk in the UK. The most recent Global Burden of Disease Report Card for the UK indicates that these are: dietary (10%), high blood pressure (8.8%), high BMI (4.7%), smoking (3.6%), physical inactivity (3.4%) i.e. potentially 30.5% of the burden of circulatory disease may be amenable to prevention through modification of just these five factors. The sole focus in the Healthy eating intentions section on consumption of five portions of fruit and vegetables is highly unlikely to capture adequately the potential influence of the NHS Health Check on dietary behaviours. Furthermore, there is no reference to the related issue of adherence to or tolerance of medication, which is known to be a challenge for many patients.

Initial Author Response:

In terms of the questionnaire falling short of evaluating specific modifiable behaviours or risk factors, the questionnaire is not inclusive of all risk factors pertaining to CVD. For instance the assessment of diet only consists of consumption of five fruits and vegetables a day. However this is the only measurable aspect of diet recorded during an NHS Health Check consultation. And as the questionnaire was made for evaluating the NHS Health Check programme, we chose to focus on quantifiable risk factors recorded during the NHS Health Check consultation. This is now made clear in the Discussion (p 16, lines 17-21). In terms of assessing other risk factors such as smoking and alcohol consumption, these were not included in the final questionnaire largely because such questions did not apply to the vast majority of respondents. Most of the 110 respondents to the 65 item questionnaire left questions pertaining to smoking unanswered. Questions pertaining to drinking had too low of a reliability for questionnaire inclusion. Hence such questions could not be included in the final questionnaire. Future studies examining populations at increased CVD risk can look into these two scales (smoking and alcohol) and how they can be incorporated into the ABCD Risk Questionnaire. During the NHS Health Check consultation, patients that are identified to be at high risk are invited for further consultations where they may be prescribed cholesterol or blood pressure lowering medication. The questionnaire in development focused primarily on the first CVD risk assessment consultation and as such did not capture information related to medication prescribing and adherence among high CVD risk patients.

Reviewer 1 Comment #3

The argument for not including all the major modifiable risk factors is not convincing.

This should be covered in the Methods section and the limitations of this choice covered in the Discussion.

This is a serious omission since both smoking and excess alcohol consumption are strong established risk factors. Even if it may not be feasible to have reliable questions about behaviours it should have been feasible to include a question relating to awareness.

But they do not necessarily take up this invitation, giving some justification for early assessment of awareness. Preconceptions are an issue for many people, particularly men, and may influence their response to the Health Check or subsequent actions.

Author Response to Comment #3:

We did not intend to be vague, and we specified our explanation in the Methods section (penultimate paragraph of the Methods section) and Discussion (paragraph 3 of the Discussion section). The authors do take the reviewer's very important point on board and have decided to reintroduce the questions in the future version of the questionnaire. We have included the following statement in paragraph 3 of Discussion: "Future studies examining populations at increased CVD risk can look into incorporating smoking and alcohol into the ABCD Risk Questionnaire."

The authors also agree with the reviewer's statement regarding the need for further assessment of people at increased risk. We clarified how individuals' level of CVD risk is assessed in paragraph 2 of Introduction.

Although subsequent invitations to those at increased CVD risk are not directly the scope of the article, the following statement is included in paragraph 2 of Introduction: "In addition to lifestyle advice given to all participants, people at high risk of CVD are invited for further consultations and offered statins and behaviour change support in relation to physical activity, smoking cessation, safe alcohol consumption and healthy diet."

Reviewer 1 Comment to Author:

There were a number of minor points that the authors may wish to address:

p6, l29/30 Criteria is the plural of criterion

p7, l36/7 Explain abbreviations CVD probably OK but HBM may not be so well known

p8, l9/10 Criteria plural again

p13, l28/9 Insert word 'be' after 'not'

p15, l38/9 'commonalities' should read 'communalities'

p17, l20/1 To substantiate 'brief' questionnaire in Conclusions, Results section should provide some indication of mean and spread of completion times

Author Response:

The terms 'criterion,' 'criteria' and 'commonalities' were correctly used. Health Belief Model is now spelled out in the first instance. The term 'brief' refers to the final number of items 26 – rather than time completion.

Reviewer 1 Comment #4

If 'criteria' is used correctly as the plural then it should be followed by 'were'. I accept that the word 'commonalities' is OK. Both terms are used in relation to factor analysis. Completion time (range) for the questionnaire would be helpful (as would the option of allowing for feedback from respondents. This point was not made in my original review but was triggered here when I considered the difficulties in responding to questions in the 26-item version of the questionnaire – included here as a suggestion for the authors to consider).

Overall, then, I do not feel that the authors have made a satisfactory response to my comments on the original version of their manuscript.

Author Response to Comment #4:

We apologise for this error. The term criteria is now followed by 'were' in all instances.

We inserted the duration it takes to complete the brief 26-item questionnaire (penultimate sentence of Methods section).

The authors hope that the reviewer finds the responses to their concerns addressed satisfactorily in this response and to all the responses above.

VERSION 3 – REVIEW

REVIEWER	Professor Tom Cochrane University of Canberra, Australia
REVIEW RETURNED	15-Jul-2017

GENERAL COMMENTS	T Reviewer 1 Comment #1 to Author: Whilst I accept that it is not possible to change the wording of the questions retrospectively, it should have been possible to get the wording right in the first place, irrespective of whether the questions are gleaned from published work or are grounded in psychology theory. Both the 85 question and the 65 question versions that preceded the reductive phase of the study, in my view, contained several questions that were worded unsatisfactorily and I believe this limitation has carried through to the proposed final 26 item version of the questionnaire. Content validity may be partially satisfactory but the wording and scoring of individual questions is not Author Response to Comment #1: The reviewer makes a valid point regarding perhaps that some of the items noted in previous responses can be interpreted as unclear or could have been worded differently, and their points are taken. The author would like to reassure the reviewer that we have scrutinised the wording of the initial questionnaire with focus group members and expert panellists which incorporated discussion and responses pertaining to the clarity, appropriateness of information presented, the amount of information presented, biases, format of question and presentation of the information using wording directly from the NHS Health Check Personal Reports (documents given directly to NHS Health Check participants. Not once were the above issues raised by members of either group in the validation process. However, it should be noted that all items above identified by the
--

reviewer are no longer presented in the brief 26-item questionnaire. We do recognise that no questionnaire can worded perfectly for every respondent and that we aim to continue to review the psychometric properties of this questionnaire in future studies with larger sample sizes. As such, we now acknowledge these realities as limitations of our work.

Reviewer 1 response to Authors reply to Comment #1

I appreciate that the authors now accept these fundamental limitations in their approach to questionnaire development. I would be uncomfortable with the Conclusions section of the Abstract that states 'The final questionnaire, with satisfactory reliability and validity, is recommended for use in assessing patients' awareness of CVD risk among NHS Health Check attendees.' I do not believe it is suitable for such use yet from content, reliability and completeness perspectives.

Reviewer 1 Comment #2

The factor structure was derived from a relatively small (110 patients), weakly representative sample using a set of 65 questions, many of which were poorly worded or ambiguous. The authors appear to accept this process with blind faith given that they argue that they see no need to alter the 26 item final version of their questionnaire. I do not consider this a satisfactory response to my comments. Questions 17-26 contain questions that may have alternative interpretations or alternative divergent scoring options. Take Q23 for example:

Increasing my exercise to at least 2 hours a week will decrease my chances of having a heart attack or stroke.

A respondent who never exercises to this extent may interpret this question as not applying to me and therefore score 0 or N/A. On the other hand, they could equally decide to Strongly agree, score 1, because they agree that this would be good even though they have no intention of adopting this behaviour or Strongly disagree, score 4, because they don't believe that this action would be good for them at all or is not feasible for them under their circumstances. Thus, there are alternative scores dependent only on the respondent's interpretation of the question.

Including negative versions of the same basic question and then reverse coding may be acceptable as a validity check but will distort scoring since these items are effectively included twice. Scoring in any case will be complicated by the inclusion of a large number of N/A options that are difficult to interpret.

Author Response to Comment #2:

Whilst the authors agree with some of the points noted above, we would like to point out that the example provided by the reviewer is a prime example of rewording previous items based on the feedback received by the focus group members of the initial validation process. It was recommended by members of this group to reword the following from: "Increasing my exercise for 30 minutes a day will decrease my chances of having a heart attack or stroke," which is the wording used from the NHS Health Check Participant Information Leaflets to "Increasing my exercise to at least 2½ hours a week will decrease my chances of having a heart attack or stroke." Going forward will keep what the reviewer has mentioned in mind and consider in future versions of the questionnaire. It should also be noted that the respondents of this questionnaire also complete measures that check their actual risk factor indicators which this

questionnaire are compared to for feedback purposes.

Reviewer 1 response to Authors reply to Comment #2
I think this response merely emphasises my point about the fundamental limitations in the approach to the development of the questionnaire.

Reviewer 1 Comment #3
The argument for not including all the major modifiable risk factors is not convincing.
This should be covered in the Methods section and the limitations of this choice covered in the Discussion.
This is a serious omission since both smoking and excess alcohol consumption are strong established risk factors. Even if it may not be feasible to have reliable questions about behaviours it should have been feasible to include a question relating to awareness. But they do not necessarily take up this invitation, giving some justification for early assessment of awareness. Preconceptions are an issue for many people, particularly men, and may influence their response to the Health Check or subsequent actions.
This issue is not addressed in the authors' response.

Author Response to Reviewer 1 Comment #3:
We did not intend to be vague, and we specified our explanation in the Methods section (penultimate paragraph of the Methods section) and Discussion (paragraph 3 of the Discussion section). The authors do take the reviewer's very important point on board and have decided to reintroduce the questions in the future version of the questionnaire. We have included the following statement in paragraph 3 of Discussion: "Future studies examining populations at increased CVD risk can look into incorporating smoking and alcohol into the ABCD Risk Questionnaire."
The authors also agree with the reviewer's statement regarding the need for further assessment of people at increased risk. We clarified how individuals' level of CVD risk is assessed in paragraph 2 of Introduction. Although subsequent invitations to those at increased CVD risk are not directly the scope of the article, the following statement is included in paragraph 2 of Introduction: "In addition to lifestyle advice given to all participants, people at high risk of CVD are invited for further consultations and offered statins and behaviour change support in relation to physical activity, smoking cessation, safe alcohol consumption and healthy diet."

Reviewer 1 Comment to Author:
There were a number of minor points that the authors may wish to address:
p6, l29/30 Criteria is the plural of criterion
p7, l36/7 Explain abbreviations CVD probably OK but HBM may not be so well known
p8, l9/10 Criteria plural again
p13, l28/9 Insert word 'be' after 'not'
p15, l38/9 'commonalities' should read 'communalities'
p17, l20/1 To substantiate 'brief' questionnaire in Conclusions, Results section should provide some indication of mean and spread of completion times

Author Response:
The terms 'criterion,' 'criteria' and 'commonalities' were correctly used. Health Belief Model is now spelled out in the first instance. The term 'brief' refers to the final number of items 26 – rather than time completion.

Reviewer 1 Comment #4
If 'criteria' is used correctly as the plural then it should be followed by 'were'. I accept that the word 'commonalities' is OK. Both terms are used in relation to factor analysis. Completion time (range) for the

	questionnaire would be helpful (as would the option of allowing for feedback from respondents. This point was not made in my original review but was triggered here when I considered the difficulties in responding to questions in the 26-item version of the questionnaire – included here as a suggestion for the authors to consider). Overall, then, I do not feel that the authors have made a satisfactory response to my comments on the original version of their manuscript. Author Response to Comment #4: We apologise for this error. The term criteria is now followed by 'were' in all instances. We inserted the duration it takes to complete the brief 26-item questionnaire (penultimate sentence of Methods section). The authors hope that the reviewer finds the responses to their concerns addressed satisfactorily in this response and to all the responses above. Reviewer 1 response to Authors reply to Comments #3 and #4 My point "But they do not necessarily take up this invitation, giving some justification for early assessment of awareness. Preconceptions are an issue for many people, particularly men, and may influence their response to the Health Check or subsequent actions." The issue of preconceptions is not addressed in the authors' response. The authors have made a satisfactory response to the remaining issues.
--	---

VERSION 3 – AUTHOR RESPONSE

Reviewer: 1

Reviewer Name: Professor Tom Cochrane

Institution and Country: University of Canberra, Australia

Please state any competing interests or state 'None declared': None declared

Please leave your comments for the authors below

Reviewer 1 Comment #1 to Author:

Whilst I accept that it is not possible to change the wording of the questions retrospectively, it should have been possible to get the wording right in the first place, irrespective of whether the questions are gleaned from published work or are grounded in psychology theory. Both the 85 question and the 65 question versions that preceded the reductive phase of the study, in my view, contained several questions that were worded unsatisfactorily and I believe this limitation has carried through to the proposed final 26 item version of the questionnaire. Content validity may be partially satisfactory but the wording and scoring of individual questions is not

Author Response to Comment #1:

The reviewer makes a valid point regarding perhaps that some of the items noted in previous responses can be interpreted as unclear or could have been worded differently, and their points are taken. The author would like to reassure the reviewer that we have scrutinised the wording of the initial questionnaire with focus group members and expert panellists which incorporated discussion and responses pertaining to the clarity, appropriateness of information presented, the amount of information presented, biases, format of question and presentation of the information using wording directly from the NHS Health Check Personal Reports (documents given directly to NHS Health Check participants). Not once were the above issues raised by members of either group in the validation process. However, it should be noted that all items above identified by the reviewer are no longer presented in the brief 26-item questionnaire. We do recognise that no questionnaire can be worded perfectly for every respondent and that we aim to continue to review the psychometric properties of this questionnaire in future studies with larger sample sizes. As such, we now acknowledge these realities as limitations of our work.

Reviewer 1 response to Authors reply to Comment #1

I appreciate that the authors now accept these fundamental limitations in their approach to questionnaire development. I would be uncomfortable with the Conclusions section of the Abstract that states 'The final questionnaire, with satisfactory reliability and validity, is recommended for use in assessing patients' awareness of CVD risk among NHS Health Check attendees.' I do not believe it is suitable for such use yet from content, reliability and completeness perspectives.

Latest Author Response to Comment #1

While we agree that the questionnaire may be amended in the future to reflect possible changes in the NHS Health Check programme such as targeting people at high CVD risk, in its present form it may be used for assessing CVD risk awareness among all programme attendees. As perceived CVD risk was positively correlated with predicted 10 year CVD risk, this only strengthens the above argument for the use of the questionnaire in assessing patients' awareness of CVD risk. Nonetheless we agree that no work is ever final, and hence the term 'final questionnaire' was changed to either 'ABCD Risk Questionnaire' or 'resulting questionnaire' throughout the manuscript. The conclusion was changed to "The resulting questionnaire, with satisfactory reliability and validity, may be used in assessing patients' awareness of CVD risk among NHS Health Check attendees."

Reviewer 1 Comment #2

The factor structure was derived from a relatively small (110 patients), weakly representative sample using a set of 65 questions, many of which were poorly worded or ambiguous. The authors appear to accept this process with blind faith given that they argue that they see no need to alter the 26 item final version of their questionnaire. I do not consider this a satisfactory response to my comments. Questions 17-26 contain questions that may have alternative interpretations or alternative divergent scoring options. Take Q23 for example:

Increasing my exercise to at least 2 hours a week will decrease my chances of having a heart attack or stroke.

A respondent who never exercises to this extent may interpret this question as not applying to me and therefore score 0 or N/A. On the other hand, they could equally decide to Strongly agree, score 1, because they agree that this would be good even though they have no intention of adopting this behaviour or Strongly disagree, score 4, because they don't believe that this action would be good for them at all or is not feasible for them under their circumstances. Thus, there are alternative scores dependent only on the respondent's interpretation of the question.

Including negative versions of the same basic question and then reverse coding may be acceptable as a validity check but will distort scoring since these items are effectively included twice. Scoring in any case will be complicated by the inclusion of a large number of N/A options that are difficult to interpret.

Author Response to Comment #2:

Whilst the authors agree with some of the points noted above, we would like to point out that the example provided by the reviewer is a prime example of rewording previous items based on the feedback received by the focus group members of the initial validation process. It was recommended by members of this group to reword the following from: "Increasing my exercise for 30 minutes a day will decrease my chances of having a heart attack or stroke," which is the wording used from the NHS Health Check Participant Information Leaflets to "Increasing my exercise to at least 2½ hours a week will decrease my chances of having a heart attack or stroke." Going forward will keep what the reviewer has mentioned in mind and consider in future versions of the questionnaire. It should also be noted that the respondents of this questionnaire also complete measures that check their actual risk factor indicators which this questionnaire are compared to for feedback purposes.

Reviewer 1 response to Authors reply to Comment #2

I think this response merely emphasises my point about the fundamental limitations in the approach to the development of the questionnaire.

Latest Author Response to Comment #2

We believe that describing the methods of questionnaire development is important for understanding how the questionnaire was derived. The point above regarding changes to the wording of the questionnaire following the patient focus group illustrates authors taking into account patient perspectives. This in turn could only strengthen the questionnaire and as such does not constitute a limitation.

Reviewer 1 Comment #3

The argument for not including all the major modifiable risk factors is not convincing.

This should be covered in the Methods section and the limitations of this choice covered in the Discussion.

This is a serious omission since both smoking and excess alcohol consumption are strong established risk factors. Even if it may not be feasible to have reliable questions about behaviours it should have been feasible to include a question relating to awareness.

But they do not necessarily take up this invitation, giving some justification for early assessment of awareness. Preconceptions are an issue for many people, particularly men, and may influence their response to the Health Check or subsequent actions.

This issue is not addressed in the authors' response.

Author Response to Reviewer 1 Comment #3:

We did not intend to be vague, and we specified our explanation in the Methods section (penultimate paragraph of the Methods section) and Discussion (paragraph 3 of the Discussion section). The authors do take the reviewer's very important point on board and have decided to reintroduce the questions in the future version of the questionnaire. We have included the following statement in paragraph 3 of Discussion: "Future studies examining populations at increased CVD risk can look into incorporating smoking and alcohol into the ABCD Risk Questionnaire."

The authors also agree with the reviewer's statement regarding the need for further assessment of people at increased risk. We clarified how individuals' level of CVD risk is assessed in paragraph 2 of Introduction. Although subsequent invitations to those at increased CVD risk are not directly the scope of the article, the following statement is included in paragraph 2 of Introduction: "In addition to lifestyle advice given to all participants, people at high risk of CVD are invited for further consultations and offered statins and behaviour change support in relation to physical activity, smoking cessation, safe alcohol consumption and healthy diet."

Reviewer 1 Comment to Author:

There were a number of minor points that the authors may wish to address:

p6, l29/30 Criteria is the plural of criterion

p7, l36/7 Explain abbreviations CVD probably OK but HBM may not be so well known

p8, l9/10 Criteria plural again

p13, l28/9 Insert word 'be' after 'not'

p15, l38/9 'commonalities' should read 'communalities'

p17, l20/1 To substantiate 'brief' questionnaire in Conclusions, Results section should provide some indication of mean and spread of completion times

Author Response:

The terms 'criterion,' 'criteria' and 'commonalities' were correctly used. Health Belief Model is now spelled out in the first instance. The term 'brief' refers to the final number of items 26 – rather than time completion.

Reviewer 1 Comment #4

If 'criteria' is used correctly as the plural then it should be followed by 'were'. I accept that the word 'commonalities' is OK. Both terms are used in relation to factor analysis. Completion time (range) for

the questionnaire would be helpful (as would the option of allowing for feedback from respondents. This point was not made in my original review but was triggered here when I considered the difficulties in responding to questions in the 26-item version of the questionnaire – included here as a suggestion for the authors to consider).

Overall, then, I do not feel that the authors have made a satisfactory response to my comments on the original version of their manuscript.

Author Response to Comment #4:

We apologise for this error. The term criteria is now followed by 'were' in all instances.

We inserted the duration it takes to complete the brief 26-item questionnaire (penultimate sentence of Methods section).

The authors hope that the reviewer finds the responses to their concerns addressed satisfactorily in this response and to all the responses above.

Reviewer 1 response to Authors reply to Comments #3 and #4

My point "But they do not necessarily take up this invitation, giving some justification for early assessment of awareness. Preconceptions are an issue for many people, particularly men, and may influence their response to the Health Check or subsequent actions."

The issue of preconceptions is not addressed in the authors' response.

The authors have made a satisfactory response to the remaining issues.

Latest Author Response to Comments #3 and #4

The link between high CVD risk and attendance of follow up care is outside of the scope of this manuscript since the population that the instrument was tested with only attended the initial NHS Health Check consultation. Nonetheless the Discussion now has a sentence in the last line of paragraph 3 stating: "Future studies examining populations at increased CVD risk can look into incorporating smoking and alcohol into the ABCD Risk Questionnaire to learn about these individuals' preconceptions and attendance of follow up care."

Thank you for reviewing our manuscript. We hope that you will find this version worthy of publication.

VERSION 4 – REVIEW

REVIEWER	Professor Tom Cochrane University of Canberra, Australia
REVIEW RETURNED	23-Aug-2017

GENERAL COMMENTS	The authors have made satisfactory alterations to their manuscript in response to my comments on their third revision.
--